# Effect of Hydroxyurea on Morphology, Proliferation, and Protein Expression on *Taenia crassiceps* WFU Strain

**DOI:** 10.3390/ijms25116061

**Published:** 2024-05-31

**Authors:** Diana G. Rios-Valencia, Karel Estrada, Arturo Calderón-Gallegos, Rocío Tirado-Mendoza, Raúl J. Bobes, Juan P. Laclette, Margarita Cabrera-Bravo

**Affiliations:** 1Department of Microbiology and Parasitology, School of Medicine, Universidad Nacional Autónoma de México, Coyoacan, Mexico City 04510, Mexico; riosdg@iibiomedicas.unam.mx (D.G.R.-V.); rtirado@facmed.unam.mx (R.T.-M.); 2Unit for Massive Sequencing and Bioinformatics, Biotechnology Institute, Universidad Nacional Autónoma de México, Coyoacan, Mexico City 04510, Mexico; karel.estrada@ibt.unam.mx; 3Department of Immunology, Biomedical Research Institute, Universidad Nacional Autónoma de México, Coyoacán, Mexico City 04510, Mexico; arturocga47@iibiomedicas.unam.mx (A.C.-G.); rbobes@iibiomedicas.unam.mx (R.J.B.)

**Keywords:** *Taenia crassiceps*, cysticercus, hydroxyurea, proliferative cells, proteome

## Abstract

Flatworms are known for their remarkable regenerative ability, one which depends on totipotent cells known as germinative cells in cestodes. Depletion of germinative cells with hydroxyurea (HU) affects the regeneration of the parasite. Here, we studied the reduction and recovery of germinative cells in *T. crassiceps* cysticerci after HU treatment (25 mM and 40 mM of HU for 6 days) through in vitro assays. Viability and morphological changes were evaluated. The recovery of cysticerci’s mobility and morphology was evaluated at 3 and 6 days, after 6 days of treatment. The number of proliferative cells was evaluated using EdU. Our results show morphological changes in the size, shape, and number of evaginated cysticerci at the 40 mM dose. The mobility of cysticerci was lower after 6 days of HU treatment at both concentrations. On days 3 and 6 of recovery after 25 mM of HU treatment, a partial recovery of the proliferative cells was observed. Proteomic and Gene Ontology analyses identified modifications in protein groups related to DNA binding, DNA damage, glycolytic enzymes, cytoskeleton, skeletal muscle, and RNA binding.

## 1. Introduction

*Taenia crassiceps* is a tapeworm that commonly infects foxes and dogs, with intermediate hosts including rats, mice, and other small rodents [1]. In humans, *T. crassiceps* infections are very rare, but isolated cases have been described in immunosuppressed individuals [2,3,4,5]. *T. crassiceps* has been used as model for the study of human and porcine cysticercosis produced by *Taenia solium* [6], particularly the genetics of the host’s susceptibility to infection and the host’s immune response; this has addressed a number of questions related to the host–parasite interplay and several aspects of diagnosis, vaccination, and drug treatment [7,8,9,10,11]. In particular, this animal model has allowed the exploration of new avenues of drug treatments, considering the current limitations of conventional drugs such as praziquantel, niclosamide, or albendazole [12]. Furthermore, as the genome of *T. crassiceps* is now available [13], it opens new avenues for studies of comparative genomics between both tapeworms.

Regeneration is known as the process in which some organisms re-form (partially or completely) a missing body part; this phenomenon has been described in several metazoans and involves a number of molecular and cellular mechanisms, including identification of cell precursors, differentiation, and morphogenesis [14,15]. In flatworms, complete regeneration processes occur in planarians; it has been observed that they can regenerate any part of their body from a fragment of tissue [16,17]. In cestodes, a distinct part of the body (head and neck) can be regenerated [18]. Additionally, cell suspensions obtained from larvae of *T. crassiceps* can regenerate the complete cysticerci [19], emphasizing the role of stem cells in the regeneration process [14]. Studies have aimed to identify and understand the roles of these totipotent cell populations in the context of the capacity for self-renewal and differentiation; known as neoblasts, they display qualities similar to those of other stem cells [20,21,22,23]. Furthermore, the generation of newly differentiated cells depends exclusively on the proliferation and differentiation of this cell lineage [18,23,24].

The morphology of neoblasts has been described in planarians and acoelians: they are small cells (~10 μm), round to ovoid, with a high nucleus/cytoplasm ratio, and a cytoplasm with a few mitochondria, numerous free ribosomes, and scarce organelles [25,26]. Neoblast-like cells have been analyzed in planarians such as *Schmidtea mediterranea* and *Dugesia japonica* [27,28] and in trematodes, such as *Schistosoma mansoni* [23]. These cells play several important roles during the life cycle, particularly in the modulation of infectivity and reproduction [23].

In cestodes, different studies have identified a population of undifferentiated stem cells similar to the neoblast; known as germinative cells, they are involved in the parasite’s development and the transitions between its different life stages [29]. Moreover, they are the only cells that are mitotically active, self-renewing, and capable of differentiating into any type of somatic cell [30,31]. The biological functions of germinative cells are not well-understood, and their study is complicated due to the complex life-cycle of cestodes, which includes at least two hosts [32]. In *Echinococcus multilocularis*, germinative cells have been described as fusiform or pear-shaped, measuring 5–12 μm, and as having a large, round nucleus and 1–3 nucleoli [30].

The molecular and structural characterization of germinative cells has also been carried out, including the use of several strategies to inhibit these cells, such as irradiation or hydroxyurea (HU) treatment for the inhibition of DNA replication [29,33]. HU treatment arrests the S phase in proliferative cells, delaying mitosis and halting cell cycle progression until DNA replication is complete [34]. HU is also commonly used, in other areas of research, for the synchronization of cell lines [35]. However, when used in prolonged treatments or at high doses, HU can cause cell death due to the accumulation of DNA damage, oxidative stress, and genotoxicity [36,37].

HU treatment and irradiation have been used to study germinative cells in the free-living flatworm *Macrostomum lignano* [38], as well as in cestodes such as *Hymenolepis diminuta* [18] and *E. granulosus* [30]. Studies on adult *H. diminuta* worms showed that when germinative cells are depleted, the regeneration of adult worms stops [18]. Furthermore, the use of proliferation markers, such as phosphorylated histone 3 and EdU (5-ethynyl-2′-deoxyuriine), have been used to characterize these cells in different organisms [18,30].

In *T. crassiceps*, isolated cells can regenerate into cysticerci when injected into the peritoneal cavity of host mice [19]. Interest has grown in understanding the functionality of germinative cells in this parasite species because of its convenient in vitro culture and reproduction in laboratory mice [6]. In this context, the identification of proliferation markers and the isolation of germinative cells could allow the study of the latter’s role in the physiology of this parasite, as well as the production of transfected organisms. Here, we report the effect of HU on *T. crassiceps* proliferative cells, including the parasite’s ability to recover after HU treatment. Our proteomic studies identified a number of alterations in the expression of metabolic proteins and cytoskeletal proteins induced by HU treatment, opening the possibility of identifying new therapeutic drug targets. Our hypothesis is that we can reduce the regenerative potential of cysticerci through reducing the number of proliferative cells by HU treatment. The research goal of this study was to characterize the effects of this treatment through a proteomic analysis, as a first effort to identify how sensitive proliferative cells are to this highly toxic treatment, as well as to initiate the evaluation of their potential and their possible roles in the survival, metabolism, and other physiological aspects of the larvae.

## 2. Results

### 2.1. HU Produced Several Physiological Changes on T. crassiceps Cysticerci Maintained In Vitro

The morphology, mobility, and metabolic activity of the cysticerci, as well as the integrity of the tegument, were analyzed after HU treatment (for 6 days) at concentrations of 25 mM and 40 mM. When groups of ten larvae were incubated with HU at a concentration of 25 mM, the main change was a decrease in the motility of the cysticerci (Appendix A). Parasites treated with 40 mM of HU showed reduced size compared to the controls, with some of them becoming swollen and some others evincing elongation changes (Appendix A). The mobility in this group decreased significantly, with only a few larvae (1–3) per group showing slow movements after treatment. In both HU-treated groups the tendency to evaginate increased, even after the HU treatment was suspended. The percent of evagination in the 25 mM groups was around 30% and reached up to 50% in the 40 mM groups, in contrast to the control groups, where evagination remained negligible at the end of the in vitro culture (Figure 1A). When the 25 mM HU treatment was suspended and the cysticerci were maintained in culture medium for recovery, the larvae appeared to increase motility, whereas no improvement was apparent after 40 mM treatment (Appendix A).

Cysticerci treated with both concentrations of HU for 6 days showed a significant decrease in metabolic activity (evaluated by the biotransformation of Alamar Blue) compared to untreated (control) cysticerci (Figure 1B). This difference did not change after three days of recovery; however, after six days of recovery, all groups, including the control, showed a reduction in metabolic activity, which was more notorious under the HU 40 mM treatment.

Tegument integrity was evaluated with the fluorescent marker Sytox Green. Cysticerci treated with 25 mM of HU showed similar intensity of the fluorescence marker when compared with the control group. In the case of cysticerci treated with 40 mM of HU, a higher fluorescent intensity was observed (Appendix A). However, the significance level of these assays was difficult to confirm.

### 2.2. Reduction in the Number of Proliferative Cells

Proliferative cells were assessed by counting the number of EdU+ cells in the tissue of the cysticerci after HU treatment. A significant decrease in the number of proliferative cells was observed after HU treatment. The number of proliferative cells decreased by approximately 60% in the 25 mM HU group, and by 90% in the 40 mM HU group (Figure 2A).

Considering recent reports showing that germinative cells can recover their proliferation capacity after HU treatment [29], we decided to evaluate the recovery of proliferative cells maintained under in vitro culture, three and six days after HU treatment. To evaluate the number of proliferative cells, areas of about 1 mm^2^ were revised in triplicates (see Section 4). The results indicated that the number of proliferative cells did not recover, but showed a gradual decrease during the three and six days that the cysticerci were maintained under in vitro culture (Figure 1B), Moreover, the number of proliferative cells also decreased in the untreated groups, indicating that conditions for in vitro culture are far from optimal for the maintenance of these cells (Figure 1B). It is worth mentioning that individual cysticerci occasionally showed a clear surge in the number of proliferative cells when visualized through EDU incorporation, suggesting the existence of diversity of responses in this clonal population of larvae.

In a previous report, we described those proliferative cells, as identified by the incorporation of EdU, in *T. crassiceps* WFU cysticerci; they were mainly localized in the nascent buds and the scolex [13]. The areas positive for proliferative cells sharply decreased in both the bladder walls (Figure 3A) and the scolices (Figure 3B) of *T. crassiceps* WFU cysticerci after 6 days of treatment with 25 mM and 40 mM of HU. However, after both 3 and 6 days of recovery, the number of proliferative cells increased, particularly in the case of the 25 mM HU group.

### 2.3. Effects of HU on Protein Expression in T. crassiceps WFU Cysticerci

Proteomic analyses were carried out on protein extracts of HU-treated and control larvae using label-free, MS-based proteomics. We identified 1062 proteins, which were then quantified to estimate changes in abundance/expression. Among them, 929 proteins were shared among the three groups (control, 25 mM HU, and 40 mM HU). Interestingly, only two proteins were exclusive to the control group, (Appendix A), corresponding to E3 ubiquitin ligase, which promotes protein ubiquitination and degradation [39], and ANK6 which plays a central role in male–female gamete recognition [40]. Five other proteins were exclusive to the 25 mM HU group (Figure 4A): leucyl aminopeptidase, which catalyzes the removal of N-terminal amino acid residues (preferentially leucine) from proteins and peptides [41]; papilin, which is involved in embryonic development and shows relation with matrix extracellular components [42]; NF1, which is related to RNA polymerase III transcription initiation and gene expression [43]; TRAB2β, which participates in the control of pre-mRNA splicing [44]; and one protein without identity. Eleven proteins were exclusive to the 40 mM HU group (Figure 4A), some of which are recognized as characteristic of the parasite, such as the major egg antigen [45]. Others were related to nucleolar machinery such as PDK1, which is involved in cell growth and proliferation [46]; NAA50, which enables H4 histone acetyltransferase activity [47]; MCTS 1, an oncogene involved in cell cycle progress [48]; molecular scaffold cullin 1; the enzyme uridine-cytidine kinase 2; the guanine nucleotide exchange factor synembrin; and the enzyme CDP-diacylglycerol synthase.

After 3 days of recovery, two proteins were found to be exclusive to the 25 mM HU group: the oxysterol-binding protein, which transports and regulates the metabolism of sterols and phospholipids [49]; and the Venom allergen protein, considered relevant in host–parasite interactions [50]. In the 40 mM HU group, four proteins were exclusive: the enzyme adenylosuccinate lyase, which participates in the purine nucleotide cycle; one DNA2 nuclease/helicase, involved in multiple DNA metabolic pathways [51]; a T-cell immunomodulatory protein; and a conserved protein (Figure 4B). The proteins exclusive to the control group are shown in Appendix A.

After 6 days of recovery, the largest numbers of independent proteins for each group were found (Figure 4C). In the 25 mM HU group, seven proteins were found: N-cadherin, involved in neural tissues; ATP-dependent zinc metalloprotease, involved in protein quality control and regulation; ribonucleoside diphosphate reductase subunit; glutathione synthetase; aconitate hydratase; mitochondrial and endoplasmic Ero1, involved in protein folding in the endoplasmic reticulum [52]; and one expressed conserved protein.

In the 40 mM HU group, ten proteins were found: the integral membrane glycoprotein prominin, the oxysterol-binding protein, involved in lipid metabolism [53]; the secretory carrier-associated membrane protein SCAMPs [54]; the nucleoporin seh1 A, which plays a role in nucleocytoplasmic transport and cell cycle regulation [55]; the Zw10 protein, which ensures proper chromosome segregation during cell division [56,57]; the ribosomal protein S23 (RPS23); and the splicing factor 3b subunit 2 (SF3B2), among others (Appendix A).

### 2.4. Differential Expression of Proteins

Our proteomic analysis identified the proteins with statistically significant changes of expression after exposing the cysticerci to HU. In the case of cysticerci treated with 25 mM of HU, 39 proteins changed (Appendix A), 21 of which increased their expression, with endophilin B1, a multifunctional protein involved in apoptosis and mitochondrial function [58], presenting the largest increase. Other proteins that increased their expression were the cytoskeletal proteins actin A3, actin type 5, actin modulator protein, filamin, and transgelin (Figure 5A). Some glycolytic enzymes also increased (fructose-biphosphate aldolase and triosphosphate isomerase). Moreover, 18 proteins decreased, mainly those concerning skeletal muscle components such as myosin and paramyosin (Figure 5A); the L18a and S3 ribosomal proteins; and the cytoskeletal proteins actin, dynein heavy chain, and talin; as well as enzymes such as α-1, 4-glucan phosphorylase, ADP/ATP translocase, and glycogen debranching enzyme, among others.

In cysticerci treated with 40 mM of HU, 21 proteins changed, 12 of which were upregulated and were mainly related to the extracellular matrix, troponin, which is implicated in skeletal muscle, and the cytoskeletal protein lamin, part of the intermediate filaments. Also, the protein A subunit, which is involved in DNA metabolism and processes such as replication; and the heat shock 70 kDa protein 4, which is a cytosolic chaperone that facilitates protein folding, degradation, complex assembly, and translocation [59]; as well as other proteins, were upregulated (Appendix A). In contrast, nine proteins presented decreased expression (Figure 5B): the fragile site associated protein C terminal, which is an indicator of DNA damage that links cell-cycle checkpoint to DNA repair pathways [60]; the scaffold protein 14-3-3; the glycolytic enzyme fructose-bisphosphate aldolase; and cytoskeletal proteins such as actin and tubulins, among other proteins (Appendix A).

In cysticerci treated with HU at 25 mM and after 3 days of recovery, a total of 38 proteins changed; 24 proteins increased their expression (Figure 5C), including two myosins and one paramyosin, which showed the largest changes, as well as the Na/K-transporting ATPase subunit alpha, the sarcomeric protein titin, the cytoskeletal kinesin-like, protein, the iron storage protein ferritin, the mixed lineage leukemia protein (mll), and histone-H3 lysine-4 (H3K4) methyltransferase. Furthermore, 14 proteins presented decreased expression compared to the control; among them were the cytoskeletal proteins actin and dynein light chain, and the glycolytic enzymes glyceraldehyde-3-phosphate dehydrogenase (GAPDH), fructose bisphosphate aldolase, and enolase (Figure 5C). The complete list of proteins is shown in Appendix A.

Treatment with HU at 40 mM produced changes in 34 proteins: 25 were upregulated, including cytoskeletal proteins such as dynein, kinesin, and other actin-binding proteins such as transgelin, profilin and myosin; nucleoside diphosphate kinase; cell division control protein 42 homolog (Cdc42), which plays a role in cellular processes including cell proliferation and migration [61]; and other metabolic enzymes, such as peptidyl-prolyl cis-trans isomerase, glutamate synthase (NADH), etc. (Appendix A). Nine proteins were downregulated, including cytoplasmic actin type 5, unc 80 protein, which is involved in monoatomic cation channel activity, as well as glycolytic and Krebs cycle enzymes (Appendix A).

As for treatment with HU at 25 mM after 6 days of recovery, 100 proteins presented modified expression. Among them, 34 increased their expression: myosin, which presented the largest increase, and cytoskeletal proteins such as tubulin beta chains, and the motor protein dynein heavy chain. Others were RNA helicase DDX15, involved in RNA splicing and ribosome biogenesis [62]; the eif4A key element, involved in the cap-dependent translation initiation process [63]; and one G1-to-S phase transition helicase orthologue of GSPT1, which has an important role in cell survival and cell cycle progression [64]. Other groups with increased expression were enzymes related to glycogen synthesis (Appendix A). Moreover, 66 proteins presented decreased expression (Figure 5E): endophilin B1, which showed the largest decrease; glycolytic and Krebs cycle enzymes and the cytoskeletal proteins dynein light chain; actin modulator protein; and oncosphere proteins tso22 and Ts8B1, among others (Appendix A).

In the 40 mM HU group after 6 days of recovery, 88 proteins changed their expression. Among them, 36 were upregulated, with the highest scores corresponding to myosin and the ribonucleotide protein major vault (Figure 5F). Other identified proteins that increased their expression included dgcr14, a transcriptional co-regulator of CD4+ T cells; the RNA helicase DDX16; and glycolytic-related enzymes, including glucose-6-phosphate isomerase, glycogen synthase kinase 3, fructose-bisphosphatase, and glycogen debranching enzyme. On the other hand, 52 proteins were downregulated, including cytoskeletal proteins and the 14-3-3 protein. Moreover, some protein expression changes coincided with those of the 25 mM dose: oncosphere tso22 and Ts8B1, as well as major egg antigen p40 (see above). Finally, thioredoxin and annexin, a Ca^2+^ and phospholipid-binding protein, reduced their expression.

### 2.5. Gene Ontology (GO) Enrichment Analysis

GO analysis identified proteins included in 97 classes related to biological process (BP) in the samples corresponding to treatment with HU at 25 mM (all classes are presented in Appendix A). Only the five representative classes are shown in Figure 6A. Concerning the ontology of cellular components (CC), the analysis identified a diverse group of proteins which are classified in 77 classes, among which mitochondrial issues, COPI vesicles coat, and integral component of membranes were the most represented classes. Finally, within the ontology of molecular function (MF), 71 classes were found, principally including proteins associated with nucleotide and GTP binding or GTPase activity related to ATP metabolic processes, mRNA processes, and spermatogenesis, as well as DNA and proliferation, DNA repair, cellular response to DNA damage stimulus, and meiotic cell cycle (Figure 6A).

The samples of the cysticerci treated with HU at 40 mM obtained 33 classes related to BP, 25 corresponding to CC, and 34 pertaining to MF (Appendix A). The five main classes for each ontology are shown in Figure 6B. Proteins included in these classes involved DNA binding, RNA binding, nucleotide binding, MAPK kinase activity, and extracellular matrix processes.

As for the HU-treated cysticerci after 3 days of recovery, we found 29 classes for BP, 26 for CC, and 23 for MC (Appendix A). The classes with the largest number of identified proteins were intracellular protein transport, maternal placenta development, and cytoskeleton process (Figure 7A). Many other classes were documented, for example, the oxidation–reduction process, ATP-driven cation transmembrane transport, and apoptotic processes (Appendix A). As for the 40 mM HU group after 3 days of recovery, the GO analysis showed 39 classes for BP, 32 for CC, and 44 for MC (Appendix A). The classes with the largest number of identified proteins were intracellular protein transport, maternal placenta development, and cytoskeleton process (Figure 7B). Other classes were cell differentiation, gluconeogenesis and other metabolic and processes. (All protein classes after 25 mM and 40 mM HU treatment and 3 days of recovery are included in Appendix A).

Finally, the 25 mM HU group after 6 days of recovery resulted in 99 classes for BP, 78 for CC, and 87 for MF (Appendix A). The identified proteins corresponded to diverse classes, including ATP binding, nucleotide binding, RNA binding, protein binding, actin binding, cysteine-type endopeptidase activity, and thiol-dependent ubiquitin-specific protease activity (Figure 7C). As for the 40 mM HU group after 6 days of recovery, the GO analysis showed the most diverse results: we identified 203 classes for BP, 136 for CC, and 135 for MC (Appendix A). The classes with the largest number of identified proteins were ATP binding, cytoskeleton, development, and protein kinase activity.

## 3. Discussion

In the present study, we established that HU is a useful tool for the study of proliferative cells in *T. crassiceps* cysticerci, similar to other cestode models such as *M. ligano*, *H. diminuta*, *M. corti*, and *E. granulosus* [18,30,33,38]. Moreover, a parasiticide effect was observed in cysticerci treated with HU at a concentration of 40 mM, which correlates with the decrease in metabolic activity as indicated by Alamar Blue and Sytox green assays. Partial EdU incorporation has been correlated with modified DNA synthesis leading to cell death in *E. granulosus* [65]. In this context, a parasiticide effect similar to that of praziquantel has been reported after in vivo and in vitro single exposure to HU in the trematode *S. mansoni*, with the principal consequences being the damage of the tegument, vacuolization of the subtegumental cells, and disorganization of muscle layers [66]. These findings are consistent with the morphological changes observed here in *T. crassiceps* cysticerci treated with HU (40 mM), particularly the disruption of the tegument in the in vitro culture and the parasiticidal effect of HU, as well as the protein expression changes identified in the proteomic analysis, which indicate an altered expression of cytoskeleton and skeletal muscle proteins.

HU treatment results in the inhibition of DNA replication, affecting the activity of ribonucleotide reductase leading to the induction of a replication stress [67,68]. The cytotoxic effects of HU are usually connected to two events: the accumulation of DNA damage and the induction of reactive oxygen species (ROS) [36]. Moreover, high concentrations or long exposure to HU compromises DNA integrity and can even generate chromosome damage [37,69]. In our experiments with *T. crassiceps* cysticerci treated with HU at a concentration of 25 mM, GO analysis demonstrated changes that are consistent with a physiological response to DNA damage: DNA duplex unwinding, DNA repair, DNA double strand-break, etc.; these are usually associated with the effects of HU on ribonucleotide reductase over DNA strands and chromosomes [67]. Of particular interest were changes in the expression of ribonucleoside diphosphate reductase, which are other main targets of HU [37,70]. Additionally, some protein classes identified through GO analysis, such as age-dependent response to oxidative stress, cellular oxidant-detoxification, and cell redox, were also identified [36].

In the *T. crassiceps* cysticerci, untreated proliferative cells are enriched in the nascent buds and the scolex, which has been reported before [13]. The analysis of cell inhibition showed that HU treatment (25 and 40 mM) resulted in the depletion of proliferative cells in developing larvae. This strong reduction generated a wide range of physiological changes in the parasite, affecting vital functions such as cell cycle, morphogenesis, development, and cell differentiation, among others. It has been reported that other processes, for instance, spermatogenesis, are also affected by HU and can induce testicular germinative cell apoptosis in mice [71]. In a closely related organism, the tapeworm *M. lignano*, a reduction in the expression of the vasa orthologue, a well-known marker of germinative cells in the testes of HU-treated animals, is indicative of alterations in the germinative cells and testis of the parasite [38]. Additional studies are needed to identify, assess, and evaluate changes in the expression of markers of neoblasts or germinative cells in *T. crassiceps*, such as polo-like kinase 1 (PLK1) in *E. granulosus* and *T. solium* [72,73], the protein nanos in *E. granulosus* [30], PL10/DDX3 in *M. corti* [33], the mini-chromosome maintenance 2 (mcm-2) in *H. diminuta* [18], and vasa in *M. lignano* [39].

The proliferative cells of *T. crassiceps* cysticerci presented a partial recovery after both 3 and 6 days with an absence of HU. This was particularly evident in the case of HU treatment at a concentration of 25 mM, similar to what is reported in *E. granulosus* [30]. The recovery of proliferative cells in nascent buds was not clear in the larvae that were treated with 40 mM HU. This recovery could be related to the increased expression of proteins involved in proliferation, development, and the cell cycle (transcription factor FAR1 protein mll) on day 3 of recovery, and the G1-to-S phase transition protein, RNA helicase DDX15, and two eif4A proteins on day 6 of recovery.

Proteomic analysis in flatworms can been used as a strategy to identify proteins with a role in the host–parasite relationship, pathogenesis, immune evasion, etc., as candidates for vaccine development, diagnostic biomarkers, or new drug targets [74,75]. This rationale has been widely used in other organisms, including as *Trypanosoma cruzi* [76,77] and *Leishmania donovani* [78]. In the case of tapeworms, this is an emergent research area; for example, in a recent study seeking to improve mefloquine design as a treatment of *E. multilocularis*, the 25 most abundant binding proteins with participation in energy metabolism, cellular transport, structure, and nucleic acid binding were selected between 1681 potential protein targets for mefloquine and its derivatives [79].

Proteomic studies are also an important tool for the characterization of HU mechanism of action; the analysis indicated a decrease in the expression of metabolic proteins mainly related to glycolysis, gluconeogenesis, and lipid biosynthesis, as well as an increase in the expression of cytoskeletal proteins. This might be relevant, since alterations in the expression of cytoskeletal proteins, like actin and tubulin, and many proteins related to actin, e.g., ankyrin, have been documented after exposure to HU [80,81]. Another group of proteins with altered expressions after HU treatment were glycolytic enzymes such as GAPDH, fructose-bisphosphate aldolase, and enolase [81].

At present, our inference of the physiological and metabolic responses of *T. crassiceps* cysticerci to HU treatment is still sketchy and limited. Future studies will effect analyses of more specific aspects, in order to refine the detail of the responses. For example, transcriptomic analysis will be useful in identifying specific genes related to germinative cells and the regeneration phenomenon. Moreover, the use of genetic strategies that allow analysis of specific cell linages, as well as the gene manipulation techniques (transfection, silencing or modification) involved in proliferative cells, could help understand the potential and the mechanisms of *T. crassiceps* regeneration at a cellular and molecular level. Finally, analysis of germinative cells in a model organism facilitates the evaluation of regeneration and differentiation in these organisms, which might aid in the understanding of equivalent phenomena in humans, specifically, those involved in tumor proliferation and other events associated with cancer [82].

## 4. Materials and Methods

### 4.1. Supply of T. crassiceps Cysticerci

Cysticerci were propagated through consecutive intraperitoneal passages of cysticerci, from female infected BALB/c mice to naïve mice, as described before [6]. After 9–12 weeks of infection, larvae were recovered from the peritonea of mice under sterile conditions and were subsequently washed with 1× PBS (pH 7.4) supplemented with 4% antibiotic/antimycotic (GIBCO 15240-062). Subsequently, 10 cysticerci/well were placed in 12-well culture plates with 2 mL RPMI medium supplemented with 1% antibiotic/antimycotic. All experiments with mice included in this study were approved by the Institutional Committee for the Care and Use of Laboratory Animals (CICUAL) at the School of Medicine (permission number 001-CIC-2022).

### 4.2. HU Treatment of T. crassiceps Cysticerci

The groups of 10 larvae maintained were treated for 6 days with HU in the culture medium, at concentrations of 25 mM and 40 mM, and at 37 °C under an atmosphere of 5% CO_2_. The medium with HU was replaced every day for 6 consecutive days. Cysticerci were allowed to recover after the treatment with HU and were transferred to fresh culture medium supplemented with 10% fetal bovine serum and antibiotic/antimycotic for in vitro culture for 3 or 6 days. Plates were revised daily to observe the mobility and morphology of cysticerci.

### 4.3. Effects of HU on Whole Cysticerci

Changes in cysticerci’s shape were evaluated by triplicates, under a Motic SMZ-171-TLED Stereo Microscope. The evaluation of cysticerci motility was made following categories devised to estimated changes in the motility of the cysticerci: (+ + + completely motile), (+ + mildly motile), (+ slightly motile), and (0 non-motile) [83]. To determine tegument integrity, 5 µM of SYTOX Green Nucleic Acid Stain (S7020; Thermo-Fisher Scientific, Waltham, MA, USA) was added to the culture medium for 30 min, following the instructions of the manufacturer. Then, cysticerci were fixed in 4% paraformaldehyde in PBS and observed under an inverted Olympus IX71 microscope at low magnification (10×) using a 488 ± 15 nm wavelength for excitation. The metabolic activity of cysticerci was assessed by biotransformation using the Alamar Blue^®^ reagent (Invitrogen Life Technologies, Carlsbad, CA, USA). Spectrophotometric readings of the recovered supernatants were recorded at Synergy™ HTX Multi-Mode Microplate Reader (BioTek, El Segundo, CA, USA), using an absorbance of 570/600 nm, and the percent reduction of Alamar Blue was calculated. The results were analyzed on Prisma using a normality test Shapiro–Wilk test, one-way ANOVA and the Dunnet test. All samples were evaluated in triplicate.

### 4.4. Identification of Proliferative Cells in Cysticerci of T. crassiceps WFU Strain

The cysticerci obtained from the mouse peritonea were incubated in RPMI-1640 medium (GIBCO™) supplemented with 10% Bovine Fetal Serum (BFS) and 1% antibiotic/antimycotic (GIBCO™). Subsequently, 5-ethinyl-2′-deoxyuridine (EdU) (Click-iT™, Life Technologies, Carlsbad, CA, USA) was used, at a final concentration of 25 µM, for 5 h, after which it was replaced with fresh medium and allowed to incubate for 24 h. The cysticerci were fixed with 4% paraformaldehyde and permeabilized with acetone (≥99.9%, Sigma, Co., St. Louis, MO, USA) for 5 min. To detect EdU, the manufacturer’s instructions were followed (Click-iT™ EdU Alexa 555, Life Technologies). To detect cell nuclei, the cysticerci were incubated with 2-(4-amidinophenyl)-1H-indole-6-carboxamidine (DAPI) (Invitrogen Life Technologies, Carlsbad, CA, USA) at a concentration of 1 µg/mL for 30 min. The cysticerci were covered with a mounting solution (40% glycerol-PBS) and observed under a confocal microscope Nikon A1R+ STORM. The quantification of proliferative cells was performed across three assays in triplicate using FIJI (ImageJ) software (https://fiji.sc/, accessed date 23 May 2024) to evaluate areas of ~1 mm^2^. The data were analyzed on Prisma software version 8.0.1 using a normality test, the Shapiro–Wilk test, two-way ANOVA, and the Dunnet test.

### 4.5. Mass Spectrometry of Protein Extracts of Cysticerci Treated with HU

The *T. crassiceps* cysticerci treated with HU and the controls were washed 3 times with sterile PBS and subsequently homogenized in solubilization buffer (7 M urea, 2 M thiourea, 4% CHAPS, 10 mM Tris pH 7.3, complemented with protease inhibitors: 12.5 mM EDTA, 1 μM pepstatin, 1 mM PMSF, and 0.1 mM leupeptin). Afterwards, the cysticerci were subjected to freezing/thawing cycles 3 times and were centrifuged at 13,800× *g* for 15 min at 4 °C. The solubilized protein was quantified using the Non-Interfering Protein Assay™ kit (G-Biosciences, St. Louis, MO, USA). The protein profile was obtained through sodium dodecyl sulfate-polyacrylamide gel electrophoresis (SDS-PAGE) under reducing conditions (2.5% 2-β-mercaptoethanol) on 10% polyacrylamide gels. The proteins were separated using the PowerPac 3000 power supply (Bio-Rad, Hercules, CA, USA).

For mass spectrometry, the crude extract was run on a 4% acrylamide/bis-acrylamide gel under reducing conditions with 2.5% 2-mercaptoethanol. A band extending beyond the 10% acrylamide/bis-acrylamide gel was stained with Bio-Safe Coomassie Blue, destained with distilled water and cut with a scalpel under sterile conditions. Mass spectrometry analysis was carried out at the Proteomics and MS Core Facility, State University of New York.

### 4.6. Bioinformatic Analysis

The MS data was searched using SequestHT in Proteome Discoverer [84] (version 2.4, Thermo Scientific, Waltham, MA, USA) against the following databases simultaneously: *E. granulosus* and human (Uniprot: https://www.uniprot.org/, accessed on 20 May 2024), and a list of common laboratory contaminant proteins (Thermo Scientific). Identification of annotated signal peptides was carried out using SignalP v4.1 [85]. Changes in protein expression and the Gene Ontology analysis were performed using specialized software, Scaffold 5 version 5.3.0.

## 5. Conclusions

One of the main results of the present study is the confirmation that HU is a useful tool for the study of proliferative cells in *T. crassiceps* larvae, similar to its use with other cestodes. When depleting germinative cells by HU treatment, several aspects of the parasite’s physiology were affected, including changes in mobility, shape, and metabolism, among others, a finding which could help to understand the role of germinative cell populations in these organisms. HU treatment of *T. crassiceps* cysticerci was highly cytotoxic on the larvae. Our proteomic analyses have determined a number of expression changes that might explain not only the proteins that are significantly down or upregulated, but also the metabolic pathways affected by specific depletion of the germinative cell population. This information could aid identification of HU protein targets during treatment of cysticerci and can be compared with other equivalent effects in related organisms. The goal is the characterization of the role of proliferative cells in the thriving of this parasite in the context of a complex host–parasite relationship.

## Figures and Tables

**Figure 1 ijms-25-06061-f001:**
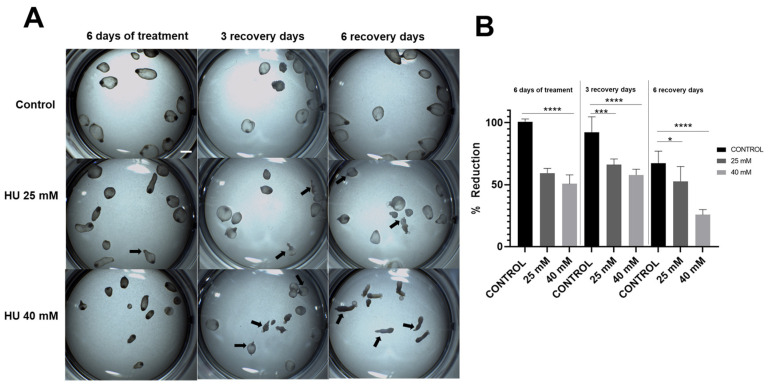
In vitro evaluation of the effect of hydroxyurea (HU) in *T. crassiceps* cysticerci. (**A**) Changes in the shape of cysticerci were detected by stereomicroscopy. Two concentrations of HU (25 mM and 40 mM) were applied for 6 days; the cysticerci were also evaluated after 3 and 6 days of recovery in HU-free medium. Black arrows show evaginated cysticerci. Scale bar represent 2 mm. (**B**) Changes in metabolic activity were measured through the reduction of Alamar Blue^®^ (Y-axis) over a period of 24 h in the different groups (X-axis). **** *p* < 0.0001, *** *p* = 0.0001, and * *p* = 0.01.

**Figure 2 ijms-25-06061-f002:**
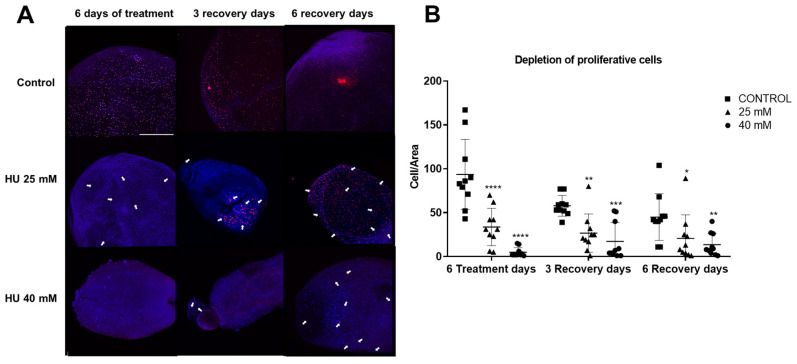
The effect of HU treatment on the number of proliferative cells in the tissue of *T. crassiceps* cysticerci. (**A**) Whole-mount localization of proliferative cells (indicated with white arrows) on the bladder wall tissue of *T. crassiceps* WFU cysticerci. Subsequently, the mounts were stained with EdU (red) and DAPI (blue) to mark proliferative cells in phase S and cell nuclei, respectively. Bars represent 200 µm; *n* = 10–12 individual larvae in all groups. (**B**) The effect of HU treatment (25 mM or 40 mM) on the number of proliferative cells in the tissue from the cysticerci. Also, the effects of 3 and 6 days of recovery in in vitro cultures are shown. Groups of larvae without HU treatment were used as controls. The statistical analysis of the results was carried out using a two-way ANOVA and the Dunnet comparison test. **** *p* < 0.0001, *** *p* = 0.0001, ** *p* = 0.001, and * *p* = 0.01.

**Figure 3 ijms-25-06061-f003:**
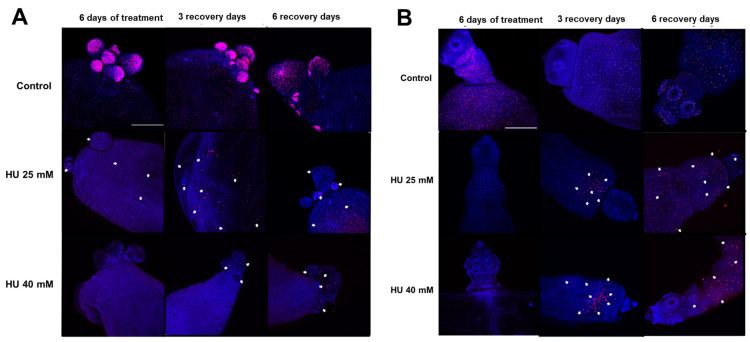
Proliferative cells in *T. crassiceps* WFU cysticerci after 6 days of HU treatment. Proliferative cells (phase S) (indicated with white arrows) were observed after staining with EdU (red); DAPI (blue) was used to counterstain cell nuclei. Here, representative pictures of whole mounts of (**A**) the bladder walls and (**B**) the scolices are shown. Bars represent 200 µm.

**Figure 4 ijms-25-06061-f004:**
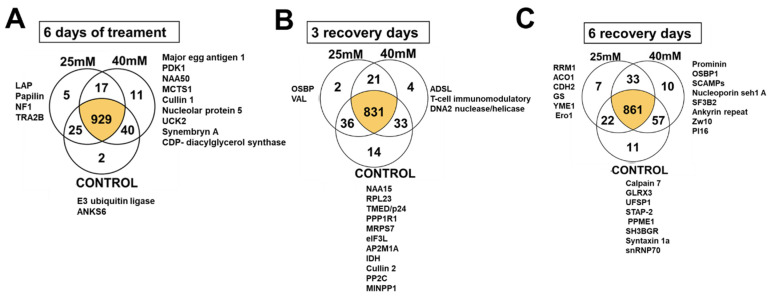
Proteins expressed after HU treatment. Venn diagram of differentially expressed proteins in different groups: (**A**) 6 days of HU treatment, (**B**) 3 days of recovery, and (**C**) 6 days of recovery. Only proteins with annotation are present in the figure.

**Figure 5 ijms-25-06061-f005:**
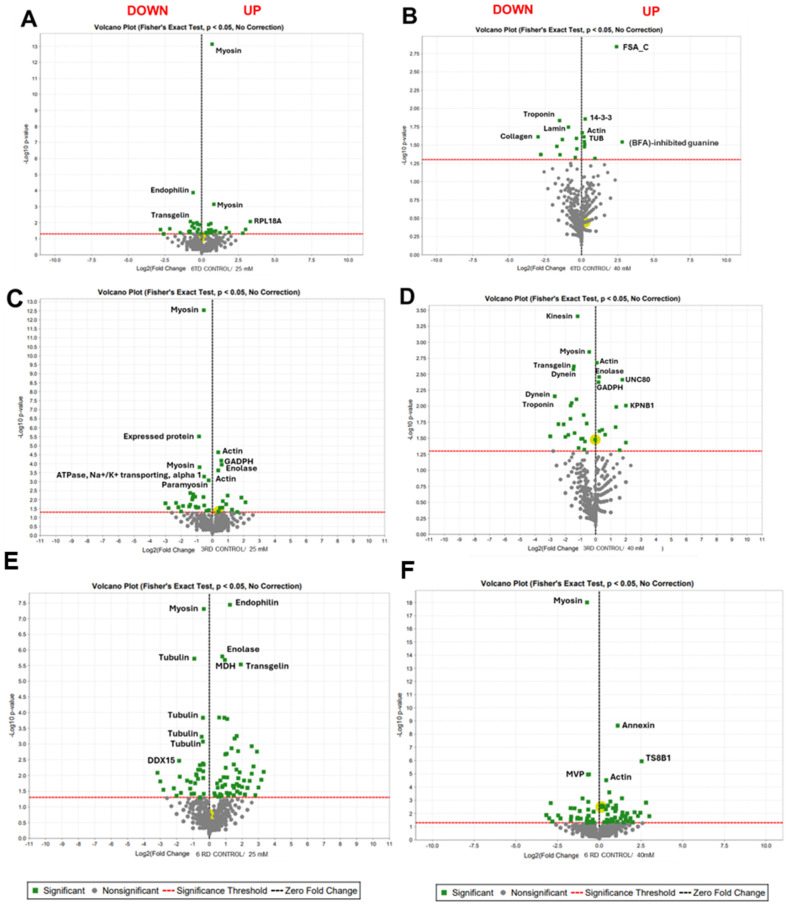
Volcano plot of differentially expressed proteins identified through mass spectrometry. The X-axis represents the log2 of expression fold change whereas the Y-axis represents -Log10 negative logarithm base 10 (-Log10) of the *p*-value. In green are indicating statistical significance values where a *p*-value ≤ 0.05. Downregulated proteins are shown on the right side of the plot (HU high/control low), whereas upregulated proteins are on the left side (HU low/control high). Non-significant changes are displayed in grey. (**A**) Control vs. HU, 25 mM, and 6 days of treatment; (**B**) Control vs. HU, 40 mM, and 6 days of treatment; (**C**) Control vs. HU, 25 mM, and 3 days of recovery; (**D**) Control vs. HU, 40 mM, and 3 days of recovery; (**E**) Control vs. HU, 25 mM, and 6 days of recovery; and (**F**) Control vs. HU, 40 mM, and 6 days of recovery.

**Figure 6 ijms-25-06061-f006:**
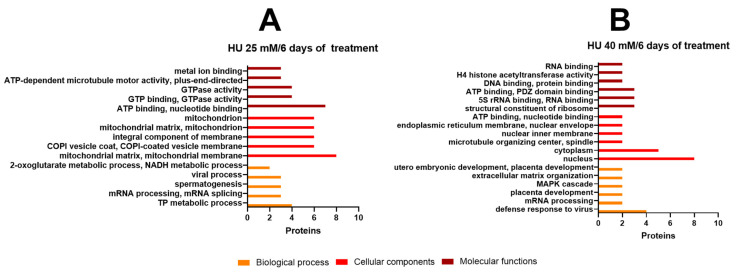
Gene ontology (GO) analysis of differentially expressed proteins after HU treatment at concentrations of (**A**) 25 mM and (**B**) 40 mM. Within the three ontologies, Biological Processes (BP): orange; Cellular Components (CC): red; Molecular Function (MF): dark red; the top five protein classes are shown. All classes are included in Appendix A.

**Figure 7 ijms-25-06061-f007:**
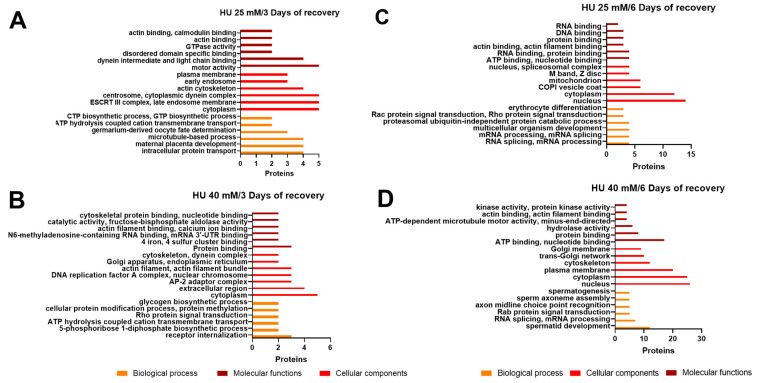
Gene ontology (GO) analyses of differentially expressed proteins after HU treatment and 3 days of recovery. Within the three ontologies (BP: orange; CC: red; and MF: dark red), the top five protein classes are shown. (**A**) HU 25 mM, 3 days of recovery; (**B**) HU 40 mM, 3 days of recovery; (**C**) HU 25 mM, 6 days of recovery; and (**D**) HU 40 mM, 6 days of recovery. All classes are included in Appendix A.

## Data Availability

The original contributions presented in the study are included in the article/Appendix A, further inquiries can be directed to the corresponding author/s.

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
