# Peer review of "Effect of Hydroxyurea on Morphology, Proliferation, and Protein Expression on *Taenia crassiceps* WFU Strain"

_ijms, 2024, doi:10.3390/ijms25116061_

Round 1

Reviewer 1 Report

Comments and Suggestions for Authors

I have the following concerns:

1. The materials and methods section should be placed after the introduction and before the results. It should also be condensed in order to contain the most important elements. 

2. Please add a novel paragraph in the discussion, analyzing the restrictions of your study in detail. 

3. In humans, HU (hydroxyurea) is known to increase HbF hemoglobin concentration and to decrease the blood cells, lowering hematocrit, white blood cells and platelets. I was wondering if any similar comparison regarding the above data or thoughts could be made using HU on Taenia crassiceps WFU Strain.

4. Please place the main findings of your study in a separate table. 

5. What are the applications of the findings of your study to i) other flatworm strains and ii) possibly to humans. Please add a novel paragraph to the discussion. 

Comments on the Quality of English Language

I suggest moderate editing of the english in the manuscript. 

Author Response

Reviewer 1

I have the following concerns:

Q: The materials and methods section should be placed after the introduction and before the results. It should also be condensed in order to contain the most important elements. 

A: The materials and methods were placed according to the template of the journal; requiring the Materials and Methods section to be placed at the end of manuscript. As for the recommendation of condensing our methods, a number of unnecessary reiterations were eliminated without scarifying clarity of the text; the number of characters of this section was reduced from 5,977 to 4,265.

Q: Please add a novel paragraph in the discussion, analyzing the restrictions of your study in detail. 

A: A paragraph analyzing the restrictions of our study was added to the las paragraph of the Discussion (lines 392-403)

Q: In humans, HU (hydroxyurea) is known to increase HbF hemoglobin concentration and to decrease the blood cells, lowering hematocrit, white blood cells and platelets. I was wondering if any similar comparison regarding the above data or thoughts could be made using HU on Taenia crassiceps WFU strain.

A: The reviewer’s mention of the HU effect to increase HbF and to lower hematocrit, white blood cells and platelets in humans is highly suggestive and interesting, considering our previous studies  showing that T. solium is able to uptake and possibly use hemoglobin, haptoglobin and other chaperons for the parasite’s management of iron. We will consider this insight in future studies on T. crassiceps. As for the possible effect of HU on T. crassiceps, our current results suggest a closer relationship with proteins involved in glycolysis and cytoskeleton that are also associated with blood alterations in in patients treated with HU.

Q: Please place the main findings of your study in a separate table. 

A: We have prepared a figure summarizing the main findings of our studies that could be used as a graphical abstract for our manuscript. We hope that this figure fulfils the requirement of the reviewer.

Q: What are the applications of the findings of your study to i) other flatworm strains and ii) possibly to humans. Please add a novel paragraph to the discussion. 

A: In order to fulfil the recommendation of the reviewer, in addition to the summarized findings of our work in the initial paragraphs of the Discussion, we have added a last paragraph in the same section; in fact, the only conclusion paragraph left in this revised version, is a reflection on these issues. We deeply appreciate the critical and useful reviewer’s suggestions that make our manuscript to read much better.

Reviewer 2 Report

Comments and Suggestions for Authors

The authors measured the activity of hydroxyurea (HU) against germinative cells of T. crassiceps cysticerci, an experimental model for tapeworm larvae development. Two different concentrations of the drug were used at three time points after exposure of larvae to the drug: on the 6th day of larvae culture in vitro, and then 3 and 6 days after recovery of larvae in the replaced free of HU culture medium.

The authors concluded that after exposition to HU, the larvae size, shape and motility changed. However, these results are not supported by numerical data.  HU decreased the number of totipotent cells and along the time of recovery time, the number of cells increased.

Proteomic and Gene Ontologies analyses identified modifications in protein groups related to DNA binding, DNA damage, glycolytic enzymes, cytoskeleton, skeletal muscle, and RNA binding.

General concept comments

The inhibitory activity of HU against platyhelminths germinative cells in tapeworms has already been evaluated in earlier studies (DOI: 10.1186/2041-9139-5-10). The novelty of the presented studies is providing the proteomic and gene Onthologies of identified proteins responding to the treatment.

The experimental design is appropriate to test the aim of the study. A gap in the knowledge was identified although the hypothesis is not fully expressed. The manuscript needs several improvements for scientific sound. Some names and terms require correction because they are incorrect and mean something different than what is stated. The results require the presentation of metric values, such as size (length and width of the larva) and the number of larvae with a protruding scale or changing shape per unit of time. The authors seek the proliferative markers of germinative cells and measure tapeworm physiology in vitro. It was established that parasite cells may recover after HU treatment. However many proteins were identified but specific were not indicated.

The figures need better presentation and data by statistical evaluation. Not all statements are effectively justified in the manuscript.

Specific comments 

Line 72:

Factual error in the name of the development stage of tapeworm. Please change the term cyst into cysticercus (or larvae) throughout the manuscript, as the larval stage has the shape of a cyst but is not a cyst.

Line 83

Generally speaking, as a part of sciences, physiology belongs to biology. The chapter title may be changed to something more general.

Line 88

The film shows not peristaltic movement, but pulsatory movements of the larva's integuments.

Line 98

Please provide the number of cysticercosis evaginated in each group.

Line 99

In the sentence " Changes in the morphology of cysticerci were detected by stereomicroscopy. " I think you mean changes in shape. Morphology is an anatomical term specific to a species, not to a single organism in motion.

Line 109

Please discuss the result in Figure 1B for 25 mM HU after 6 days of recovery. This does not change compared to 6 days of treatment and 3 days of recovery, while the control group had a reduced parameter.

Lines 110-113

Please specify what you mean by "tegument stability". What is measured by this fluorescence method?

Line 134

Please draw the white arrows to indicate the cells.

Line 140

Could you please provide the results of the two-way ANOVA, considering time and group as variable parameters?

Line 147

Please express "some" in numbers.

Line 152

Please draw the white arrows to indicate the cells.

Line 341

Physiological changes overlap with biological changes, so their distinction seems artificial. Use more specific terms: motility and level of physiological activity of tapeworm larvae.

Line 413

Please write clearly for how many days the larvae were exposed to HU.

In the delivered edition, the manuscript has to enhance the value and standard of the obtained results and therefore would be reconsidered after major revision.  

Author Response

Reviewer 2

Comments and Suggestions for Authors

The authors measured the activity of hydroxyurea (HU) against germinative cells of T. crassiceps cysticerci, an experimental model for tapeworm larvae development. Two different concentrations of the drug were used at three time points after exposure of larvae to the drug: on the 6th day of larvae culture in vitro, and then 3 and 6 days after recovery of larvae in the replaced free of HU culture medium.

The authors concluded that after exposition to HU, the larvae size, shape and motility changed. However, these results are not supported by numerical data.  HU decreased the number of totipotent cells and along the time of recovery time, the number of cells increased.

Proteomic and Gene Ontologies analyses identified modifications in protein groups related to DNA binding, DNA damage, glycolytic enzymes, cytoskeleton, skeletal muscle, and RNA binding.

General concept comments

The inhibitory activity of HU against platyhelminths germinative cells in tapeworms has already been evaluated in earlier studies (DOI: 10.1186/2041-9139-5-10). The novelty of the presented studies is providing the proteomic and gene ontologies of identified proteins responding to the treatment.

The experimental design is appropriate to test the aim of the study. A gap in the knowledge was identified although the hypothesis is not fully expressed. The manuscript needs several improvements for scientific sound. Some names and terms require correction because they are incorrect and mean something different than what is stated. The results require the presentation of metric values, such as size (length and width of the larva) and the number of larvae with a protruding scale or changing shape per unit of time. The authors seek the proliferative markers of germinative cells and measure tapeworm physiology in vitro. It was established that parasite cells may recover after HU treatment. However many proteins were identified but specific were not indicated.

The figures need better presentation and data by statistical evaluation. Not all statements are effectively justified in the manuscript.

A: Metric values suggested by the reviewer were added (lines 94-97), images are now provided as Supplementary Fig. 1A.

Specific comments 

Line 72:

Q: Factual error in the name of the development stage of tapeworm. Please change the term cyst into cysticercus (or larvae) throughout the manuscript, as the larval stage has the shape of a cyst but is not a cyst.

A: The reviewer is right in the sense that “cyst” is an informal term; accordingly, we have changed all mentions in the text to cysticerci/cysticercus or larvae.

Line 83

Q: Generally speaking, as a part of sciences, physiology belongs to biology. The chapter title may be changed to something more general.

A: The reviewer is right, and we have changed the title of the section to “HU produced several

physiological changes on T. crassiceps cysticerci maintained in vitro

Line 88

Q: The film shows not peristaltic movement, but pulsatory movements of the larva's integuments.

A: In order to solve the reviewer´s comment, we have decided to remove these films from the supplementary materials. Instead, we included a new Supplementary Figure 1 (see below), showing quantitative results of the cysticerci motility in treated and control groups.

This figure is now explained in the methods section 4.3: “Effect of HU on whole cysticerci” (supplementary Fig. 1A). Additionally, Paragraph 2.1 of the Results has been considerably edited to clarify the effect of HU treatment.

Line 98

Q: Please provide the number of cysticercosis evaginated in each group.

A: The percent of evaginated cysticerci under each treatment is now mentioned in paragraph 2.1 of Results.

Line 99

Q: In the sentence "Changes in the morphology of cysticerci were detected by stereomicroscopy. " I think you mean changes in shape. Morphology is an anatomical term specific to a species, not to a single organism in motion.

A: To comply with reviewer’s comment, Fig. 1 legend has been edited to: “Changes in shape of cysticerci were detected by stereomicroscopy”

Line 109

Q: Please discuss the result in Figure 1B for 25 mM HU after 6 days of recovery. This does not change compared to 6 days of treatment and 3 days of recovery, while the control group had a reduced parameter.

A: Cysticerci treated with both concentrations of HU for 6 days showed a significant decrease in metabolic activity (evaluated by the biotransformation of alamar blue) compared to untreated (Control) cysticerci (Figure 1B). This difference did not changed after three days of recovery, however, after six days of recovery, all groups, including the control, showed a reduction in metabolic activity, which was more notorious under HU 40 mM treatment. This explanation was added to the text in 116-120 lines.

Lines 110-113

Q: Please specify what you mean by "tegument stability". What is measured by this fluorescence method?

A: In these assays, we are using Sytox green which is a marker used to evaluate viability, unable to penetrate an intact membrane. In the case of our experiments, Sytox green will only stain nuclear DNA in the bladder wall nuclei, if it can penetrate the tegument. Therefore, we agree with the reviewer that “stability” is an unclear term, and have replaced it by integrity.

Line 134

Q: Please draw the white arrows to indicate the cells.

A: We have followed the reviewer’s suggestion placing a number of arrows to point cells

Line 140

Q: Could you please provide the results of the two-way ANOVA, considering time and group as variable parameters?

A: The results are now presented in a two-way ANOVA as suggested by the reviewer. We also carried out a normality test (Shapiro-Wilk) that resulted in a α= 0.05. The resulting graph, grouping all times and conditions was similar, showing gradual decreases in the number of proliferative cells in the control and in the 25 and 40 mM treated groups. This is clearly stated in the manuscript (lines 133-140)

Line 147

Q: Please express "some" in numbers.

A: We have deleted the complete sentence.

Line 152

Q: Please draw the white arrows to indicate the cells.

A: Arrows have been drawn in figures 2 and 3 to show proliferative cells, as suggested by the reviewer.

Line 341

Q: Physiological changes overlap with biological changes, so their distinction seems artificial. Use more specific terms: motility and level of physiological activity of tapeworm larvae.

A: We deleted the term “biological” to avoid the conceptual overlapping detected by the reviewer.

Line 413

Q: Please write clearly for how many days the larvae were exposed to HU.

A: We have now indicated: “Groups of 10 cysts maintained in 12-well culture plates were treated for 6 days with HU in the culture medium, at a concentration of 25 mM and 40 mM”, as suggested by the reviewer.

Q: In the delivered edition, the manuscript has to enhance the value and standard of the obtained results and therefore would be reconsidered after major revision.  

A: We appreciate careful review of our manuscript; we have considered and presented answers to all suggestions. We expect that our answers are adequate. Now the manuscript reads much better.

Reviewer 3 Report

Comments and Suggestions for Authors

Overall, the work “Effect of Hydroxyurea on morphology, proliferation, and protein expression on Taenia crassiceps WFU strain” seems thorough and well-structured. Here are some suggestions which need to incorporate before accepting the article:

It will be helpful to clearly state the research goal or hypothesis. What specific aspect of the regeneration process or reproductive cells have you studied?

When mentioning the process of cell regeneration and germination, it may be beneficial to define these terms for readers who may not be familiar with them and to address a broader audience. Adding a concise description can improve the clarity of your article.

Expand on the significance of your findings in the Discussion section. How do your findings contribute to our understanding of flatworm regeneration or the role of reproductive cells? Discuss any unexpected findings and possible implications for future research.

Conclude your article with suggestions for future research based on the study's conclusions and new research directions.

Comments on the Quality of English Language

No comments

Author Response

Reviewer 3

Comments and Suggestions for Authors

Overall, the work “Effect of Hydroxyurea on morphology, proliferation, and protein expression on Taenia crassiceps WFU strain” seems thorough and well-structured. Here are some suggestions which need to incorporate before accepting the article:

Q: It will be helpful to clearly state the research goal or hypothesis. What specific aspect of the regeneration process or reproductive cells have you studied?

A: In this study, T. crassiceps cysticerci received a treatment with HU, looking to characterize (research goal) its effects through a proteomic analysis, as a first effort to identify how sensitive are proliferative cells to this highly toxic treatment, as well as to initiate evaluation of their potential and possible role in the survival, metabolism, and other physiological aspects of the larvae. Our hypothesis being that we can reduce regenerative potential of cysticerci through reducing the number of proliferative cells by HU treatment. Future studies will be directed to study regeneration through recovery or transplantation of the proliferative cells. This is now stated at the end of the Introduction section (lines 84-89).

Q: When mentioning the process of cell regeneration and germination, it may be beneficial to define these terms for readers who may not be familiar with them and to address a broader audience. Adding a concise description can improve the clarity of your article.

A: New sentences were added to the first paragraph of the Introduction to comply with the recommendation of the reviewer.

Q: Expand on the significance of your findings in the Discussion section. How do your findings contribute to our understanding of flatworm regeneration or the role of reproductive cells? Discuss any unexpected findings and possible implications for future research.

A: In order to fulfil the recommendation of the reviewer, in addition to the summarized findings of our work in the initial paragraphs of the Discussion, we have added a last paragraph in the same section; in fact, the only conclusion paragraph left in this revised version is a reflection on these issues.

Q: Conclude your article with suggestions for future research based on the study's conclusions and new research directions.

 A: Following the recommendation of the reviewer the conclusion was greatly reduced and modified.

Round 2

Reviewer 1 Report

Comments and Suggestions for Authors

I have no further concerns. 

Author Response

Thank you very much for your pertinent and important suggestions that have aided improving the quality of our manuscript.